# Association between Infection and Calculated Globulin Level among Patients with Thymic Epithelial Tumor

**DOI:** 10.3390/jcm13185600

**Published:** 2024-09-21

**Authors:** Joyce Cui, Tawee Tanvetyanon

**Affiliations:** 1Morsani College of Medicine, University of South Florida, Tampa, FL 33602, USA; cuij@usf.edu; 2Moffitt Cancer Center, University of South Florida, Tampa, FL 33612, USA

**Keywords:** thymoma, infection, hypogammaglobulinemia, globulin, thymic carcinoma

## Abstract

**Background**: Thymic epithelial tumors (TETs) are uncommon malignancies uniquely associated with autoimmunity and immunodeficiency. Previous studies among patients with primary immunodeficiency diseases have shown that a low calculated globulin (CG) level, obtained by subtracting albumin from total protein level, is associated with infection risk. We investigated this association among patients with TET. **Methods**: A cross-sectional retrospective study was performed based on electronic medical records of patients with TET treated during 2002–2024 at a tertiary care institution. For each patient, their lowest CG level and the date of occurrence were identified. The incidence of serious infection requiring hospitalization during 6 months before and 6 months after the index date was recorded. Multivariable Poisson regression models were constructed. **Results**: Among 101 TET patients, 96 patients (95%) had the information available to derive at least one CG level. The median lowest CG level was 2.65 g/dL (range 1.0–4.2). There were 33 serious infection episodes. Pneumonia was the most prevalent type of infection in 52% of episodes. In a multivariable analysis, a CG level below 2.0 was independently associated with the prevalence of infection with a prevalence ratio of 6.18 (95% CI: 3.12–12.23, *p* < 0.001). Furthermore, thymectomy was significantly associated with infection. **Conclusions**: Among patients with TET, a low CG level was associated with an increased prevalence of serious infections. Our limited experiences suggest that it is feasible to derive the CG level for most patients during routine clinical care.

## 1. Introduction

A thymic epithelial tumor (TET) is a rare malignancy originating from the thymus, an organ essential for the maturation of T cells, which are responsible for adaptive immunity. Based on data from the Surveillance Epidemiology End Results, the incidence of TET in the United States is estimated to be about 3 per 1,000,000 persons [1]. According to the World Health Organization 2021 classification, TET can be broadly classified into thymoma type A, type AB, type B (subclassified into B1, B2, and B3), and thymic carcinoma [2]. For resectable TET, the cornerstone of treatment is surgery followed by adjuvant radiotherapy among selected cases [3]. For unresectable or distantly metastatic TET, several systemic therapy options are available.

In addition to the inherent risk of disease progression, patients with TET also face the risk of paraneoplastic autoimmune disorders. Driven by the dysfunctional immune tolerance mechanism caused by TET, autoimmune conditions, including myasthenia gravis and pure red cell aplasia, disproportionately occur in this population [4]. Furthermore, an immunodeficient state may occur due to dysfunctional T- or B-lymphocytes, leading to opportunistic infections. The clinical spectrum of thymoma-associated immunodeficiency includes Good’s syndrome and thymoma-associated immunodeficiency without hypogammaglobulinemia [5]. Good’s syndrome was first described in 1955 by Dr. Good and is characterized by low gamma globulin level (IgG) as well as impaired T-cell-mediated immunity [6]. An immunodeficient state poses a significant risk of morbidity and mortality for patients with TET. It has been estimated that 14% of deaths among patients with Good’s syndrome were due to an infection [7]. 

Unfortunately, serum IgG level, a key diagnostic parameter for Good’s syndrome, is not included in routine laboratory tests. Nonetheless, for many patients with TET, serum protein and albumin levels are measured during routine clinical care, thus allowing for the calculation of globulin levels. When the serum albumin level is subtracted from protein, a calculated globulin (CG) level can be readily derived. In some clinical settings, CG can be a useful screening test for hypogammaglobulinemia, which is associated with the risk of an opportunistic infection [8,9,10,11]. For example, in populations with primary immunodeficiency, CG levels ranging from 1.8 to 2.4 g/dL have been shown to correlate with hypogammaglobulinemia, a situation where prophylactic IgG replacement therapy is indicated. To our knowledge, however, no study on CG level has been conducted among the TET patient population. In this paper, we investigate the association between CG levels and infections.

## 2. Materials and Methods

### 2.1. Patients and Medical Record Review

This study utilized a cross-sectional, retrospective design. After approval from the Scientific Review Committee and Institutional Review Board, a list of medical records of patients with TET evaluated at our institution between 2002 and 2024 was obtained from the Tumor Registry. Patients were included in the review if they had a confirmed pathological diagnosis of TET available. Laboratory data were obtained, and CG levels were derived. The lowest CG level for each patient and the date of its occurrence were recorded to serve as an index date. The occurrence of serious infection in the period 6 months before and 6 months after the date of the lowest CG level was recorded. For patients with the lowest CG levels observed on multiple occasions, the occasion closest to the initial active treatment, defined as surgery for localized disease or chemotherapy for advanced disease, was chosen as an index date.

### 2.2. Definitions

CG level was derived by subtracting serum albumin level from total serum protein. The definition of a low CG level in the published literature varies. In this study, we defined a low CG level as <2 g/dL. We also used other cut-off levels for sensitivity analysis. The outcome of interest was serious infection, defined as an infection requiring hospitalization for over 24 h, and included episodes occurring at outside institutions. The histological classification of TET was based on the WHO and was provided as available in the medical records [2]. Comorbidity was classified according to Charlson’s index [12]. The stage at diagnosis was based on the Masaoka–Koga staging system. The use of immunosuppressive drugs, defined as corticosteroids of over 10 mg per day, including chemotherapy within 1 month of the index date, was documented. For the classification of infectious disease, microbial-type diagnoses, such as the identification of cytomegalovirus infection, were abstracted from medical records, when available, as were organ-type diagnosis such as pneumonia. Among patients who underwent thymectomy, the interval from thymectomy to the date of the first infection episode was obtained. 

### 2.3. Statistical Consideration

Descriptive statistics, including median and range for continuous variables and proportion and frequency for categorical variables, were used. Pearson’s Chi-square or Fisher’s Exact test was used as appropriate to examine the association between low CG levels and the presence of infection. Univariable and multivariable Poisson regression models were constructed to examine the association between clinical factors and the prevalence of infection. All *p*-values were two-sided and considered significant at <0.05. Analyses and graphs were created on SPSS version 24 (IBM corporation, Armonk, NY, USA). 

## 3. Results

### 3.1. Patient Characteristics

The medical records of 101 patients were screened, and among them, 96 patients had data available for calculated globulin (Table 1). Their median age was 55 years (range 29–89), and among patients undergoing thymectomy, the median pathological tumor size was 6.5 cm (range 1.2–31.0). The most common paraneoplastic condition was myasthenia gravis, which occurred in 16 patients, followed by pure red cell aplasia and immune-mediated cytopenia, which occurred in 10 patients. Lichen planus, symptoms of an inappropriate antidiuretic hormone, and Raynaud’s phenomenon occurred in one patient each. The use of immunosuppressive drugs was observed in 21 patients, with a comparable proportion among patients with higher vs. lower CG levels. These drugs were chemotherapeutic agents (cyclophosphamide, doxorubicin, cisplatin, carboplatin, paclitaxel, docetaxel) used with or without corticosteroids in nine patients, corticosteroids with or without biological agents used in four patients, and non-corticosteroids (efgartigimod, cyclosporin, azathioprine, low-dose methotrexate, anti-thymocyte globulin, rituximab) used in eight patients. 

Among 96 patients with available CG levels, the median lowest CG level was 2.65 g/dL (range 1.0–4.2) (Figure 1). When comparing those with the lower CG level <2.0 g/dL with those with the higher CG level ≥2.0 g/dL, there was a significantly higher proportion of patients with paraneoplastic syndrome among those with lower globulin levels, 58% compared with 27%. No significant difference was found between the groups in terms of age, sex, smoking, comorbidity, histological type, thymectomy, or pathological tumor size. Serum IgG measurements were available from 3 patients: 657 mg/dL, 332 mg/dL, and 300 mg/dL. Their lowest CG levels were 2.1 g/dL, 1.8 g/dL, and 2.0 g/dL, respectively.

### 3.2. Serious Infections

During our observation period, 33 episodes of serious infection were documented: 16 episodes among patients with higher CG levels and 17 episodes among those with lower CG levels (Table 2). The most prevalent infection was pneumonia which occurred in 17 episodes in total. All episodes were community-acquired, except for one episode which appeared to be hospital-acquired Serratia Marcescens pneumonia. Among patients who developed a serious infection (N = 18), the shortest hospitalization lasted 2 days, while the longest lasted 30 days, with a mean duration of 13.3 days. 

When classifying patients according to the number of infection episodes that had occurred (Figure 2), a higher proportion of patients with higher CG levels experienced no infection compared to those with lower CG levels. Specifically, among patients with higher CG levels (N = 82): 67 had no infection, 14 had one episode of infection, and 1 had two episodes of infection. Among those with lower CG levels (N = 14), 4 had no infection, 5 had one episode of infection, 3 had two episodes of infection, and 2 had 3 episodes of infection. 

### 3.3. Association between CG Level and Prevalence of Infection

In univariable analysis (Table 3), the prevalence of infection was associated with several factors, including CG level, paraneoplastic syndrome, smoking history, the use of immunosuppressive drugs, and thymectomy. In a multivariable model (Table 3), however, only CG level and thymectomy were found to be independently associated with the prevalence of infection. Patients with a CG level <2.0 and those who had thymectomy experienced a higher prevalence of infection with prevalence ratios of 6.18 (95% CI: 3.12–12.23) and 2.20 (95% CI: 1.02–4.74), respectively.

### 3.4. Sensitivity Analysis

In this patient cohort, the median value of the lowest CG level was 2.65 g/dL (Figure 3). We performed a sensitivity analysis using a different definition of a low CG level by using a median value of the lowest CG level in this study population, which was 2.65 g/dL. By this definition, there were 48 patients with lower CG levels and 48 patients with higher CG levels. Among the patients with the lower CG levels, 29 patients had no infection, 13 had one episode, 4 had two episodes, and 2 had three episodes. Among those with higher CG levels, 42 patients had no infection and 6 had one episode. In univariable analysis, a lower CG level was significantly associated with increased prevalence of infection, with a prevalence ratio of 4.50 (95% CI: 1.86–10.89, *p* < 0.001). In a multivariable analysis, after accounting for thymectomy, a lower CG level remained independently associated with the increased prevalence of infection, with a prevalence ratio of 4.73 (95% CI: 1.95–11.48, *p* = 0.001). 

### 3.5. Association between Interval from Thymectomy and Prevalence of Infection

Because thymectomy was independently associated with the prevalence of infection, we examined the relevance of the time interval from thymectomy. Among patients who had thymectomy, 20 experienced an infection. Of these, 17 patients had the infection after thymectomy. The median duration from thymectomy to infection was 98.2 months (range 4.0–231.4 months). We found no significant association between the duration and the prevalence of infection, *p* = 0.76.

## 4. Discussion

In this study, we examined the association between the CG level and the occurrence of serious infection among TET patients. We found that the prevalence of serious infection was higher among patients with lower CG levels compared to those with higher CG levels. Furthermore, paraneoplastic autoimmune conditions occurred more frequently among patients with lower CG levels. 

To our knowledge, this is the first study to investigate the association between CG levels and serious infections in the TET population. The occurrence of serious infection among patients with TET and low CG levels appeared to be in line with previous studies of patients with Good’s syndrome. In a systematic review that included 162 patients with Good’s syndrome, 92.6% of patients suffered serious infections [7]. In addition, sinopulmonary infection was the most common type of infection [7]. Hypogammaglobulinemia can directly impair the clearance of encapsulated bacterial pathogens, responsible for respiratory tract infections [13]. Since a low CG level is associated with increased infection, as observed in our study, and it has also been associated with hypogammaglobulinemia in other conditions, it is likely that a low CG level is predictive of hypogammaglobulinemia among patients with TET. It should be noted, however, that patients with TET may be at risk of infection even in the absence of hypogammaglobulinemia. For example, the presence of autoantibody against interleukin-23, a cytokine directing chronic inflammation, can lead to severe mycobacterial, bacterial, or fungal infections among patients with TET without concurrent hypogammaglobulinemia [14]. 

In our study, we utilized a definition of a low CG level as <2.00 g/dL. A recent large study including 550 adult patients demonstrated that this CG cut-off value predicted patients with IgG < 6 g/L with 83.8% sensitivity and 74.9% specificity [9]. However, we also observed a significant association between low CG levels and an increased prevalence of infection when the definition of low CG was as high as <2.65 g/dL. Additional studies will be required to define the optimal cut-off level. Beyond the sensitivity and specificity, the level of IgG can vary with the patient population under consideration. IgG level can decrease due to renal loss, such as in nephrotic syndrome, or in catabolic states caused by burns or hospitalization. Nevertheless, in a hospitalized adult population, one study has demonstrated that a CG level with a cutoff value ≤20 g/L still maintains a positive predictive value of 82.5% in identifying hypogammaglobulinemia, which is defined as IgG ≤ 5.7 g/L [15]. It should also be noted that certain medications, such as corticosteroids or rituximab, may lower IgG levels [16]. Finally, an important consideration is the outcome of IgG replacement therapy. Whether that can translate into a meaningful reduction in infection in the TET population remains unclear. A small study suggested that there may be a role of IgG replacement therapy among patients with TET and Good’s syndrome [17]. 

Much remains unknown regarding the pathogenesis of hypogammaglobulinemia in TET. One theory is that aberrant cytokines secreted by the bone marrow or thymic stromal cells in patients with TET may interfere with B cell precursor growth and differentiation. Another theory is that the T cells themselves from TET may inhibit immunoglobulin production from B cells, an implication drawn from studies of paraneoplastic phenomena [6]. 

It is important to acknowledge that our study had some limitations. First, as a retrospective study, we relied on the presence of medical records detailing hospitalization. Therefore, any undocumented hospitalizations could lead us to underestimate the prevalence of infections. This is typically so among patients who did not require further treatment after thymectomy as they underwent surgery at this institution and continued care at outside institutions. Second, our study period included the period of the COVID-19 pandemic, and this could have increased the prevalence of respiratory infections. An anecdotal report has suggested an increased severity of COVID-19 among patients with Good’s syndrome [18]. Nevertheless, we did not come across any documented cases of COVID-19 during our study period. Finally, while we found an association between low CG levels and an increased prevalence of severe infection, we are unable to extrapolate any causal relationship due to the cross-sectional study design. We also did not have IgG levels available for most patients. 

In summary, we found an association between low CG levels and the prevalence of serious infections among TET patients. This suggested the potential utilization of CG level as a screening test for hypogammaglobulinemia patients. This could be particularly useful given that CG level could be derived for 95% of patients in our study during routine clinical care. Our study highlighted the possible role of low globulin levels in the development of serious infections. Given that infections can lead to morbidity and mortality in the patient population, the possible clinical application of this information in patients with TET may include the use of prophylactic immunoglobulin replacement among affected patients. Until then, additional studies are warranted to better understand the relationship between CG levels, IgG levels, and serious infections. 

## Figures and Tables

**Figure 1 jcm-13-05600-f001:**
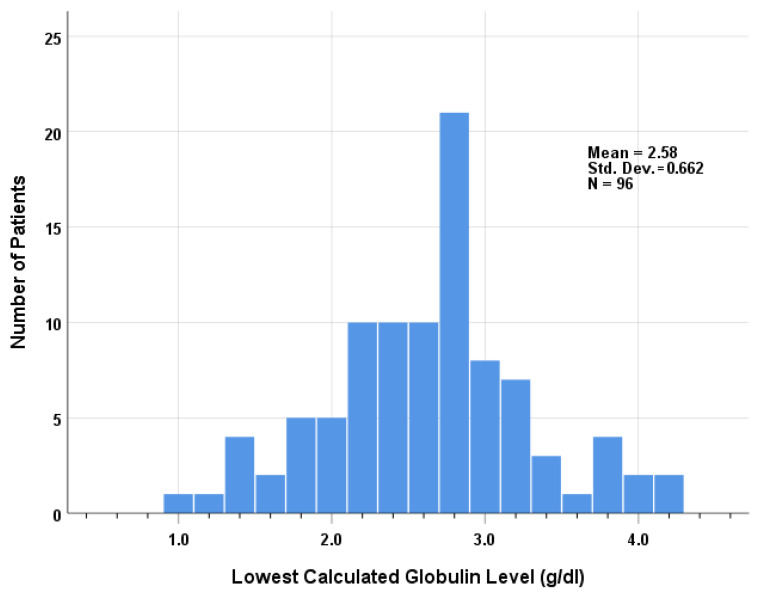
Histogram illustrating the distribution of the lowest globulin levels.

**Figure 2 jcm-13-05600-f002:**
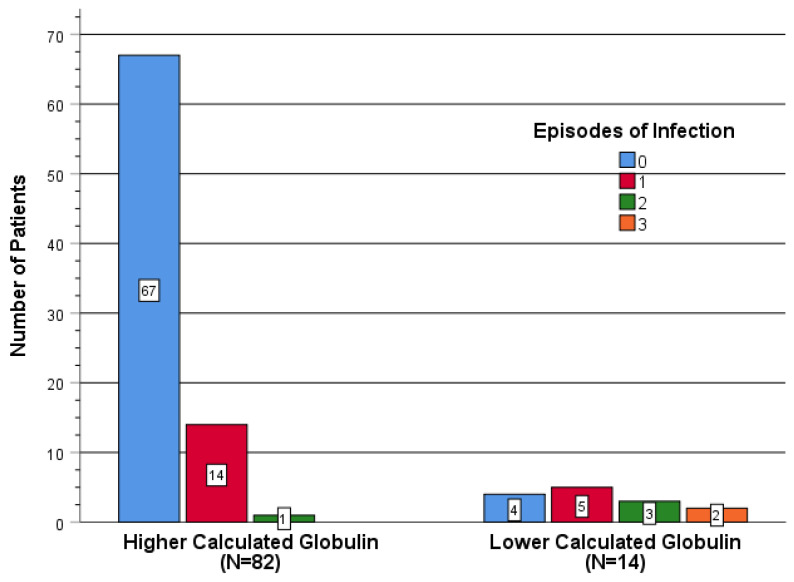
Prevalence of infection.

**Figure 3 jcm-13-05600-f003:**
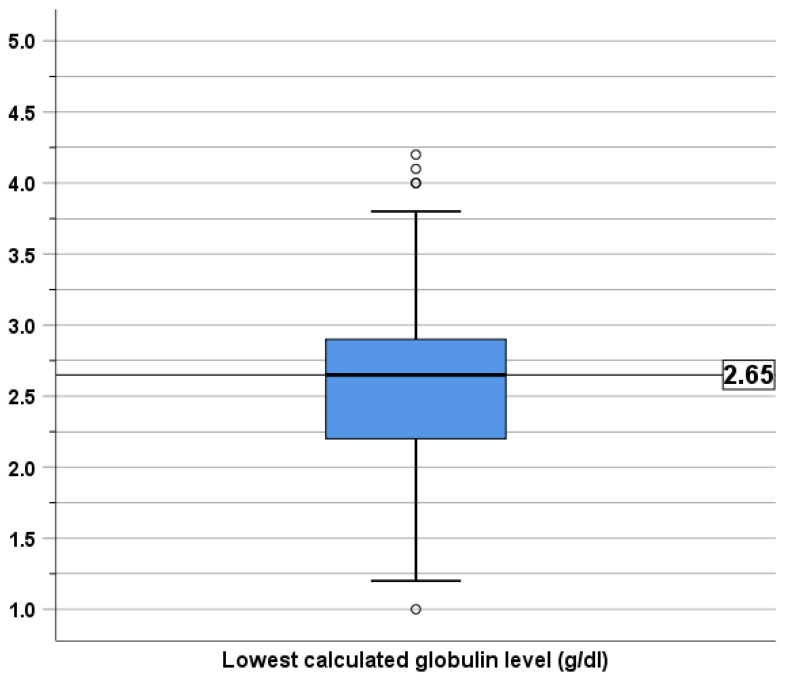
Box plot demonstrating distribution of lowest globulin levels.

**Table 1 jcm-13-05600-t001:** Patient characteristics.

Characteristics	Total Patients N = 96 (%)	Patients with Higher CG Level ≥2.0 g/dL) N = 82 (%)	Patients with Lower CG Level <2 g/dL N = 14 (%)	*p*-Value
Age:				
- <55 years	47 (49)	38 (46)	9 (64)	0.21
- ≥55 years	49 (51)	44 (54)	5 (36)	
Sex:				
- Female	47 (49)	43 (52)	4 (29)	0.09
- Male	49 (51)	39 (48)	10 (71)	
Smoking *:				
- Never smoked	45 (47)	37 (46)	8 (57)	0.56
- Current or former	50 (53)	44 (54)	6 (43)	
Comorbidity index:				
- 0	74 (77)	62 (76)	12 (86)	0.41
- ≥1	22 (23)	20 (24)	2 (14)	
Stage at diagnosis ^‡^:				
- I	20 (21)	18 (22)	2 (14)	
- IIA or IIIB	25 (26)	23 (28)	2 (14)	0.20
- III	19 (20)	15 (18)	4 (29)	
- IVA or IVB	25 (26)	19 (23)	6 (42)	
Histological type:				
- Non specified	31 (32)	26 (32)	5 (36)	
- Type A or AB	19 (20)	18 (22)	1 (7)	0.34
- Type B	40 (42)	32 (39)	8 (57)	
- Type C	6 (6)	6 (7)	0 (0)	
Thymectomy:				
- No	14 (15)	12 (15)	2 (14)	0.97
- Yes	82 (85)	70 (85)	12 (85)	
Surgical tumor size ^†^:				
- <6.5 cm	29 (52)	27 (55)	2 (29)	0.19
- ≥6.5 cm	27 (48)	22 (45)	5 (71)	
Paraneoplastic syndrome:				
- No	70 (73)	64 (78)	6 (43)	0.006
- Yes	26 (27)	18 (22)	8 (57)	
Immunosuppresive drug:				
- No	75 (78)	65 (79)	10 (71)	0.51
- Yes	21 (22)	17 (21)	4 (29)	

* data available from 95 patients; ^†^ data available from 56 patients; ^‡^ data available from 89 patients. Abbreviations: CG, calculated globulin.

**Table 2 jcm-13-05600-t002:** Characteristics of infection.

Type of Infection	Patients with Higher CG Level ≥2.0 g/dL) N = 82	Patients with Lower CG Level <2 g/dL N = 14	Total Patients N = 96
Pneumonia	8	9	17
Urinary tract infection	1	0	1
Sepsis	2	4	6
Fungal infection	1	2	3
Cellulitis	2	0	2
Viral infection (cytomegalovirus, herpes simplex virus, parvovirus	2	1	3
Pneumocystis jiroveci	0	1	1
Total	16	17	33

Abbreviations: CG, calculated globulin.

**Table 3 jcm-13-05600-t003:** Univariable and multivariable analyses of factors associated with infection.

Variable	Univariable Analysis			Multivariable Analysis		
	PR	95% CI	*p*-Value	PR	95% CI	*p*-Value
CG level:						
- <2.0 g/dL	6.22	3.14–12.32	<0.001	6.18	3.12–12.23	<0.001
- ≥2.0 g/dL	Ref	Ref		Ref		
Paraneoplastic syndrome:						
- Present	2.53	1.28–5.02	0.01	NS	NS	NS
- Absent	Ref	Ref				
Age:						
- <55 years	0.90	0.45–1.79	0.77	NS	NS	NS
- ≥55 years	Ref	Ref				
Sex:						
- Female	0.77	0.39–1.53	0.45	NS	NS	NS
- Male	Ref	Ref				
Smoking history:						
- Yes	2.22	1.08–4.58	0.03	NS	NS	NS
- No	Ref	Ref				
Charlson’s index:						
- ≥1	2.16	0.76–6.13	0.15	NS	NS	NS
- 0	Ref	Ref				
Stage at diagnosis:						
- IVA or IVB	1.42	0.69–2.93	0.34	NS	NS	NS
- Other stages	Ref	Ref				
Histology:						
- Thymic carcinoma	1.33	0.47–3.79	0.59	NS	NS	NS
- Other types	Ref	Ref				
Thymectomy:						
- Yes	2.19	1.02–4.72	0.04	2.20	1.02–4.74	0.04
- No	Ref	Ref		Ref		
Chemotherapy:						
- Yes	0.74	0.37–1.48	0.39	NS	NS	NS
- No	Ref	Ref				
Radiotherapy:						
- Yes	1.49	0.67–3.30	0.33	NS	NS	NS
- No	Ref	Ref				
Immunosuppresive drug:						
- Yes	2.04	1.00–4.15	0.04	NS	NS	NS
- No	Ref	Ref				

Abbreviations: CG, calculated globulin; PR, prevalence ratio; CI, confidence interval; Ref, reference category; NS, not significant.

## Data Availability

The original contributions presented in the study are included in the article, further inquiries can be directed to the corresponding author.

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
