# Peer review of "Association between Infection and Calculated Globulin Level among Patients with Thymic Epithelial Tumor"

_jcm, 2024, doi:10.3390/jcm13185600_

Round 1

Reviewer 1 Report

Comments and Suggestions for Authors

 In the article "Association between infection and calculated globulin level among patients with thymic epithelial tumor", the authors evaluate the possible association between low globulin levels and infection in patients with thymic tumour. The topic of this article is original and of interest to the scientific community.

I have some comments:

- according to the results of this study, patients with low globulin levels and paraneoplastic syndrome present a higher risk of infection. this group of patients is usually treated with steroids/immunosuppressive therapies. an evaluation of the possible role of steroids/immunosuppressive therapy in the development of infection in patients with paraneoplastic syndrome should be conducted. 

- The authors confirm that patients who underwent thymectomy had a higher incidence of infection. Was the time interval between surgery and the diagnosis of infection evaluated? Was antibiotic therapy given only in prophylactic doses in surgical patients?

- The discussion should be more extensive, detailing the possible role of low globulin levels in developing infection and the possible clinical application of this information in patients with thymic tumours.

Author Response

Comment 1: according to the results of this study, patients with low globulin levels and paraneoplastic syndrome present a higher risk of infection. this group of patients is usually treated with steroids/immunosuppressive therapies. an evaluation of the possible role of steroids/immunosuppressive therapy in the development of infection in patients with paraneoplastic syndrome should be conducted. 

Response 1: Thank you very much. We have collected additional data about immunosuppressive use during the time of lowest CG. We have included the data in Table 2 as well as in the result section of the text. We have also performed analysis including this new variable in the model as shown in Table 3.

Comment 2: The authors confirm that patients who underwent thymectomy had a higher incidence of infection. Was the time interval between surgery and the diagnosis of infection evaluated? Was antibiotic therapy given only in prophylactic doses in surgical patients?

Response 2: Thank you very much. We have obtained additional data regarding the timing of thymectomy and calculate the time from thymectomy to the first infection episode. We also performed an analysis and included this in the result section. We did not find a significant association. Because many patients had thymectomy many years ago, unfortunately we do not have data to ascertain the adequate use of prophylactic antibiotics.

Comment 3: The discussion should be more extensive, detailing the possible role of low globulin levels in developing infection and the possible clinical application of this information in patients with thymic tumours.

Response 3: Thank you very much. We have added additional text in the discussion section.

Reviewer 2 Report

Comments and Suggestions for Authors

In the manuscript, the authors present a study regarding the association between infection and calculated globulin level 2 among patients with thymic epithelial tumor. In my opnion, it is an interesting manuscript. In order to improve the quality of the manuscript, some changes, have to be done. My observations are : 

- the introduction part of the manuscript is too long. Some information that are presented in the introduction part of the manuscript are not related with the aim of the study.

- it is not clear in the manuscript how or why a cutoff value of 2 g/dl of the calculated globulin was chosen.

Author Response

Comments 1: the introduction part of the manuscript is too long. Some information that are presented in the introduction part of the manuscript are not related with the aim of the study.

Response 1: thank you very much. We have shortened the introduction as suggested. Due to the fact that thymoma is an uncommon malignancy and average readers may not be familiar with the natural history of disease, we have provided some background.

Comments 2: it is not clear in the manuscript how or why a cutoff value of 2 g/dl of the calculated globulin was chosen.

Response 2: Thank you very much. We have added in the discussion and included the reference. The level was chosen based on a good sensitivity as well as specificity predicting IgG<6. We have provided the reference as well.

Reviewer 3 Report

Comments and Suggestions for Authors

The manuscript presented for review is an interesting analysis of the correlation between the calculated globulin levels in patients with thymic epithelial tumors and prevalence of infection. 

The work is well organized and respects an adequate structure. My suggestions for improvement are as follows:

1. Please provide additional information on mechanisms by which thymic tumors may influence the prevalence of infection. Describe molecular pathways if possible. 

2. The study is a bit basic - there is no analysis of comorbidities, of cancer stage and histological type of tumor, nor of additional Oncologic treatment required by the patient. There is no correlation between invasive procedure which can influence infections and actual result. An analisis of community vs. hospital acquired would be adequate. Please add such information in tabelated form for ease of readability.

3. Please clarify the year's between which the study was performed - there are 2 values in abstract and in text which do not correlate.

4. Information on IgM vs. IgG should be provided when available. Similar a basal level of globulin before TET should be provided when available. This would provide additional indications of potential immunodeficiencies prior to TET which may influence results.

5 Please add data you obtained for a globin level cut-off of 2.65 as described in text. 

Author Response

Comments 1. Please provide additional information on mechanisms by which thymic tumors may influence the prevalence of infection. Describe molecular pathways if possible. 

Response 1:  Thank you very much. We have added additional text describing why thymic tumors may influence the prevalence of infection via hypogammaglobulinemia. Specifically, we explore two theories on how thymic tumors may cause antibody deficiency. 

Comments 2. The study is a bit basic - there is no analysis of comorbidities, of cancer stage and histological type of tumor, nor of additional Oncologic treatment required by the patient. There is no correlation between invasive procedure which can influence infections and actual result. An analisis of community vs. hospital acquired would be adequate. Please add such information in tabelated form for ease of readability.

Response 2: Thank you very much. We have now included information regarding Masaoka stage, included in Table 1. We have also performed analysis integrating staging information. Information on comorbidity, histological type, and oncological treatment is available as detailed in Table 1. With regard to invasive procedure such as thymectomy, we have provided additional information, as well. Most patients underwent thymectomy many years prior to the index date. With regard to hospital acquired vs. community infection, we only encountered one episode in which we can confirm it to be hospital acquired based on the identification of Serratia Marcescens. We have included this information in the result section.  

Comments 3. Please clarify the years between which the study was performed - there are 2 values in abstract and in text which do not correlate.

Response 3: Thank you for this comment. The method section has been revised. The study included patients seen between 2002-2024.

Comments 4. Information on IgM vs. IgG should be provided when available. Similar a basal level of globulin before TET should be provided when available. This would provide additional indications of potential immunodeficiencies prior to TET which may influence results.

Response 4: Thank you very much. We only have IgG level from 3 patients. We have now provided their levels in the results section. It is unknown when the patients developed TET and as such we do not have their basal globulin before the development of TET.

Comment 5: Please add data you obtained for a globin level cut-off of 2.65 as described in text. 

Response 5:  Thank you very much. We have provided Figure 3 and a description on how we derived the cut-off level of 2.65. 

Round 2

Reviewer 3 Report

Comments and Suggestions for Authors

The authors have addressed all my concerns adequately and I believe the manuscript can now be published as it is. I thank the authors for their efforts.